# A data management system for precision medicine

**John J. L. Jacobs**[1]*, **Inés Beekers**[1], **Inge Verkouter**[1], **Levi B. Richards**[1],
**Alexandra Vegelien**[1,2], **Lizan D. Bloemsma**[3], **Vera A. M. C. Bongaerts**[4], **Jacqueline Cloos**[5],
**Frederik Erkens**[6], **Patrycja Gradowska**[7], **Simon Hort**[8], **Michael Hudecek**[9],
**Manel Juan**[10,11,12], **Anke H. Maitland-van der Zee**[3], **Sergio Navarro-Velázquez**[10,11], **Lok
Lam Ngai**[5], **Qasim A. Rafiq**[13], **Carmen Sanges**[9], **Jesse Tettero**[5], **Hendrikus J. A. van Os**[4,14],
**Rimke C. Vos**[4], **Yolanda de Wit**[3], **Steven van Dijk**[1]

1 Clinical Care & Research, ORTEC B.V., Zoetermeer, The Netherlands, 2 Faculty of Mathematics, VU, Amsterdam, The Netherlands, 3 Department of Pulmonary Medicine, Amsterdam UMC, The Netherlands, 4 Public Health & Primary Care, and Health Campus The Hague, Leiden University Medical Center, The Hague, The Netherlands, 5 Department of Haematology, Amsterdam UMC, The Netherlands, 6 Department Production Metrology, Fraunhofer Institute for Production Technology IPT, Aachen, Germany, 7 HOVON Foundation, Rotterdam, The Netherlands; Department of Haematology, Erasmus MC Cancer Institute, Rotterdam, The Netherlands, 8 Adaptive Produktionssteuerung, Fraunhofer Institute for Production Technology IPT, Aachen, Germany, 9 Medizinische Klinik und Poliklinik II, University Clinic Würzburg, Würzburg, Germany, 10 Fundació Clínic per a la Recerca Biomèdica—Institut d'Investigacions Biomèdiques August Pi i Sunyer, Barcelona, Spain, 11 Immunology department, Hospital Clinic of Barcelona, Barcelona, Spain, 12 HSJD-Clinic Immunotherapy platform, Barcelona, Spain, 13 Advanced Centre for Biochemical Engineering, University College London, London, United Kingdom, 14 National eHealth Living Lab, Leiden, The Netherlands

* John.Jacobs@ortec.com

## Abstract

Precision, or personalised medicine has advanced requirements for medical data management systems (MedDMSs). MedDMS for precision medicine should be able to process hundreds of parameters from multiple sites, be adaptable while remaining in sync at multiple locations, real-time syncing to analytics and be compliant with international privacy legislation. This paper describes the LogiqSuite software solution, aimed to support a precision medicine solution at the patient care (LogiqCare), research (LogiqScience) and data science (LogiqAnalytics) level. LogiqSuite is certified and compliant with international medical data and privacy legislations. This paper evaluates a MedDMS in five types of use cases for precision medicine, ranging from data collection to algorithm development and from implementation to integration with real-world data. The MedDMS is evaluated in seven precision medicine data science projects in prehospital triage, cardiovascular disease, pulmonology, and oncology. The P4O2 consortium uses the MedDMS as an electronic case report form (eCRF) that allows real-time data management and analytics in long covid and pulmonary diseases. In an acute myeloid leukaemia, study data from different sources were integrated to facilitate easy descriptive analytics for various research questions. In the AIDPATH project, LogiqCare is used to process patient data, while LogiqScience is used for pseudonymous CAR-T cell production for cancer treatment. In both these oncological projects the data in LogiqAnalytics is also used to facilitate machine learning to develop new prediction models for clinical-decision support (CDS). The MedDMS is also evaluated for real-time

**Data Availability Statement:** All data used for conclusion data are in the manuscript and/or supporting information files. This paper also describes the collection and use of various

datasets of which the scientific conclusions, where applicable the paper refers to these papers. GDPR restricts publication of these pseudonymous data. Some data is only collected in this paper, and therefore not used for any scientific conclusions in this paper. Access to various datasets can be requested through info-nl@ortec.com.

**Funding:** The authors of this paper acknowledge funding from various sources. JJLJ, IV, IB, SH, FE, CS, SNV, MJ, CS & MH acknowledge funding from the EU framework project AIDPATH (grant agreement number 101016909). VAMCB, HJAO & RCV acknowledge funding from ZonMw (grant number 10140302110018), the Netherlands. AHMZ, LDB, & YW acknowledge funding from the PPP Allowance by Health Holland, Top Sector Life Sciences & Health (LSHM20104; LSHM20068), the Netherlands, and by Novartis. JC (institute): speakers' bureau of Astellas, research funding Takeda, DC-one, Genentech, Janssen, Novartis and Merus, royalties from BD biosciences and Navigate. LLN, PG, JT & AV have no financial disclosures to declare. The funders had no role in study design, data collection and analysis, decision to publish, or preparation of the manuscript.

**Competing interests:** We have read the journal's policy and the authors of this manuscript have the following competing interests: JJLJ, IB, IV, LBR, & SD are employees of ORTEC B.V. which has a commercial interest in LogiqSuite. AV, LDB, VAMCB, JC, FE, PG, SH, MH, MJ, AHMZ, SNV, LLN, QAR, CS, JT, HJAO, RCV & YW do not have any conflicts of interest to declare.

recording of CDS data from U-Prevent for cardiovascular risk management and from the Stroke Triage App for prehospital triage. The MedDMS is discussed in relation to other solutions for privacy-by-design, integrated data stewardship and real-time data analytics in precision medicine. LogiqSuite is used for multi-centre research study data registrations and monitoring, data analytics in interdisciplinary consortia, design of new machine learning / artificial intelligence (AI) algorithms, development of new or updated prediction models, integration of care with advanced therapy production, and real-world data monitoring in using CDS tools. The integrated MedDMS application supports data management for care and research in precision medicine.

## Author summary

Precision medicine promises more effective disease treatment by stratification and personalization of diagnosis and treatment. Further disease stratification requires more data from more patients and the use of Artificial Intelligence (AI). Traditional Medical Data Management Systems (MedDMSs) are not designed for implementation of precision medicine. We defined, built and applied a MedDMS for precision medicine. A MedDMS for precision medicine would (a) be compliant to the GDPR and other privacy protection guidelines, (b) facilitate multi-center data collaboration in research consortia, (c) allow sharing of existing and new data, (d) provide data ready for machine learning AI to develop predictive algorithms, and (e) allow the data to be used in care settings with clinical decision support tools. We developed LogiqSuite, a MedDMS compliant with these demands. We have selected use-cases from different biomedical fields, like oncology, pulmonology, cardiovascular risk management, and prehospital triage. We used LogiqSuite in multi-centre data registrations and monitoring, data analytics, for AI to develop algorithms for prediction models, to integrate care with advanced medicine production, and for real-world data monitoring.

## Introduction

Standardisation of medical practice has yielded great progress for modern medicine, but the number of new drugs approved per billion US dollars spent on research and development has halved roughly every nine years since 1950 in inflation-adjusted terms, implying that the efficacy of medical progress has gone done by a factor 80 [1–4]. Animal models of disease lack human disease variations [5,6]. Precision, or personalised medicine [7] promises significant therapeutic improvements by prediction, prevention, personalisation or stratification, and participation of patients [8,9]. It depends on distinguishing different disease mechanisms at a detail level where animal models have limited predictive value.

### Medical data management for precision medicine

Precision medicine studies clinically relevant i.e., large effects, allowing limited clinical studies [2,10] and patients stratification in small groups [11,12]. Medical data science projects increase in complexity due to the evaluation of different biomarkers, either separately or in numerous combinations [13,14]. While clinical trials in conventional medicine often limit their criteria by exclusion of potential comorbidities, precision medicine aims to include real-world data as

relevant cases of human disease [15–18]. Statistical analyses combining collections of real-world data [19,20] and clinical trial recordings [21] challenges the prerequisites for medical data management systems (medical DMSs; MedDMSs). Appropriate tools, like MedDMSs for medical data science on stratified interventions are needed for precision medicine [22,23] and lack of these lack slows down the transition to precision medicine [24]. The MedDMSs are also crucial to in data collection for the mathematical development of artificial intelligence (AI) algorithms [25].

Precision medicine defines new requirements for MedDMSs. In traditional medicine, a MedDMS has dozens of parameters from a single site. MedDMSs for precision medicine should be able to process hundreds or thousands of parameters to allow distinguishing and discrimination between various forms of disease with different mechanisms and/or requiring different interventions. It is unlikely that a single site will have relevant numbers for all stratified subgroups, thus the MedDMS should be able to integrate data from multiple sites, while maintaining compliant with privacy legislation, e.g. the EU General Data Protection Regulation (GDPR). The integration should include data validation and coordination for progressive insights.

These requirements together point towards a centrally governed cloud solution that integrates data from multiple sites, while maintaining the appropriate privacy level and access on a need-to-know basis for each subset of data.

## Requirements for a MedDMS for precision medicine

Privacy protection legislations, like the GDPR, call for privacy-by-design and fine-tuned data access [26]. Distinct medical conditions and research areas need different data models. Data models should be plastic for progressive insights and robust for long term data storage and management. Medical data are inherently complex with static data (e.g. sexes), data with a defined start and/or end date (e.g. diagnosis), repeated lab test results (e.g. haemoglobin), different kinds of sequence variations (e.g. mutations, indels, translocations), immunological cytokine profiles [27], and systems biology [28,29].

We are at the eve of implementation of precision medicine [30], but enhanced data management systems (DMSs) are needed for precision medicine [31,32]. Real-world data are important both for initial model development, but also for real-world feedback in the process of continuous learning for continual improvement of the models. Most drugs and other interventions lack data for real-world evidence, beyond the controlled clinical trials, as they are hard to capture in current medical practice [19]. Clinical studies are mostly performed under ideal circumstances with patients that fit the inclusion and exclusion criteria, which often implying a single disease without comorbidities, while real patients mostly have multiple comorbidities. Real-world data without tight inclusion and exclusion criteria increase the complexity of data.

When data become more complex, statistical rules request more data, making data sharing crucial for precision medicine. The GDPR allows the use of anonymous data in research, and pseudonymous data when patients give their consent. Medical data should be sharable between studies, implying that data should be findable, accessible, interoperable, and reusable (FAIR) [33,34]. Data dictionaries allow FAIR data dictionaries to be added to the data in LogiqSuite. In some European countries, patients should give informed consent to every scientific study that is performed with their data. Based on either opt-in or opt-out, it should be possible to include or exclude data from analytics.

Data quality is the heart of data science [35]. A developed mathematical prediction model is unlikely to be of better quality than its data input. Handling big and complex medical data are

challenges for MedDMSs and thus for implementation of precision medicine [36–39]. Data quality starts with input validation [40]. The MedDMS should also have dynamic forms to minimise redundant question [41]. It should have reports for user feedback, be integrated in care settings for real-world data, as well as research settings for trial data. Moreover continuous improvement requires data integration with clinical-decision support (CDS) tools for precision medicine.

In medical practice and research settings, responsible physicians, and investigators, respectively are assigned to cases, but their departments are also involved. Medical specialists often have authorised crosstalk between different cases of their patients, with the regulated rights for data viewing authorisation at the case level to avoid unnecessary complexity.

## Validation of the MedDMS for precision medicine

From these requirements, we set up LogiqSuite, a MedDMS to facilitate precision medicine in multicentre research studies, data analytics, AI development of prediction models, integration of care and research, and real-world data monitoring in using CDS tools. We describe five different use cases of LogiqSuite in seven topics of oncology, cardiovascular medicine, pulmonology, and pre-hospital triage. The use cases included fusing datasets, study monitoring, integrating care and research, development, and the implementation of a MedDMS. In different use cases, the usability of LogiqSuite as a MedDMS for precision medicine is evaluated.

## Materials & methods

### Concept

Our MedDMS is an integrated solution dubbed LogiqSuite, consisting of LOGIQ applications built on logic to LOG data Intelligently and Quantitatively. The concept is to record medical case data at the source to enable medical data science for precision medicine at the levels of patients, subjects, and data analytics. LogiqCare uses directly identifiable patient data to avoid patients being mix-up in clinical care. LogiqScience uses pseudonymised subject data for scientific research, laboratory, and other activities, where sample identity is crucial but direct identification of patients is undesirable. LogiqSuite allows seamless integration of LogiqCare and LogiqScience to aid collaboration between physicians and researchers. The data of LogiqCare and LogiqScience are synced to LogiqAnalytics for data analytics, in real-time, while the data are deidentified and depersonalised by controlling and minimising identifiable information. Separating LogiqAnalytics from the LogiqCare and LogiqScience database allows to filter out personal identifiers as well as to filter data not that should not be used for analytics (e.g., when records with unreliable or unconsented data) or data that do not conform to the need-to-know basis for analytics.

### Generic Design

All data handling and storage is encrypted in LogiqSuite according to the GDPR, ISO27001, and Dutch healthcare regulatory guidelines i.e., NEN7510 [42]. LogiqSuite is a robust cloud-native solution in Microsoft Azure cloud. Data are restricted to be handled and stored in the European Economic Area.

The solutions have a web interface accessible with current versions of Google Chrome, Mozilla Firefox, Microsoft Edge, and (limited) on Safari for iOS and macOS. LogiqSuite is built using modern software development best practices (continuous integration and delivery, using a test, pilot, and production deployment, automated tests, various checks, and balances

on quality). New developments are delivered under a feature toggle which allows for quickly providing new functionality or security fixes.

Data isolation can be achieved by running in a multitenant environment (logical separation) or by having a dedicated environment per project (physical isolation). Backups are done automatically for both point-in-time restore (seven days into past) and long-term full backups (weekly are retained for 30 days, monthly for 365 days). Scalability is possible in the compute, data, and web layers to cope with any workload. Specific details can be found in the technical documentation [43].

## Technical details

LogiqSuite uses standard protocols: REST API for querying data and data ingress (migration), CSV/JSON serialisation for import/export of data, TLS 1.2 for encryption of communication channels, and OIDC or SAML for integrations with Identity Providers (e.g. Active Directory) (Fig 1). Full details can be found at the technical paper [44].

Data are stored in LogiqSuite as a stream of immutable events, which allows historical reconstruction. These events are communicated over a shared message bus that any internal service can subscribe to. For example, the service that is responsible for the web site maps the events to a database that is optimised for querying data. A service from LogiqAnalytics can take the same events and map them to a database specially tailored for a specific use case. Notably, sensitive personal identifiable information (PII) can be removed during this mapping, ensuring that research data are always anonymous. All data streams in LogiqSuite are evaluated in real-time and maintain continually in sync.

LogiqSuite is developed according to privacy-by-design principles and data sharing on a need-to know basis, using the cloud-native capabilities of Azure (App Services, Azure SQL, Cosmos DB, Vault, Service Bus, etc). LogiqSuite relies on Auth0 for authentication-as-a-service [45] i.e., to establish connections with customer identity providers in a reliable and secure manner. Customers can also connect using just their own account, without the need for an enterprise connection. Multifactor authentication (MFA) is enforced for individual users.

LogiqAnalytics leverages the Microsoft Power BI platform to allow users to collaborate on datasets and create insightful dashboards. Users of the same project can share dashboards using the Microsoft Teams environment.

The Sciencrew platform allows publishing data dictionaries for integration with LogiqSuite. LogiqSuite is continuously improved [46]. LogiqCare & LogiqScience provide a set of REST APIs for integration with third party systems for data accessibility [47].

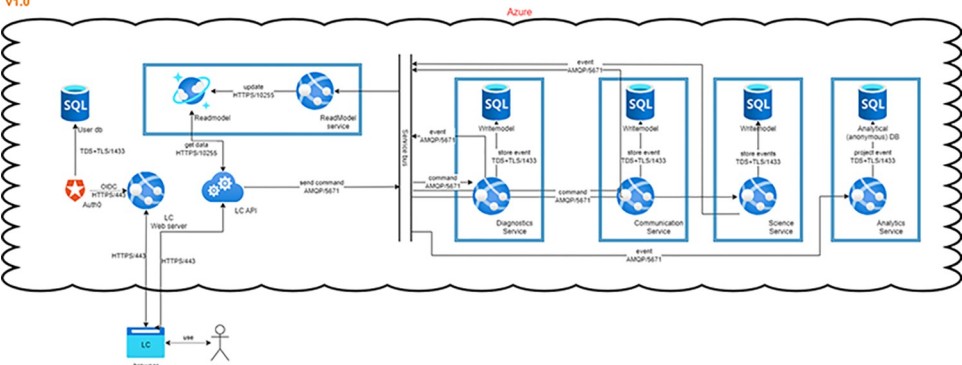

**Fig 1. Data flows in LogiqSuite.** In brief, data are stored in read-, and write-model data bases, where the write-model is the single source of true, and the read a fast cache of aggregated data.

### Case-centred data in LogiqCare and LogiqScience

LogiqCare and LogiqScience provide different access to case data of patients and subjects, respectively. Depending on the users' roles, directly identifiable or pseudonymous data are hidden, viewable, or editable. Depending on the implementation use case, patients, and subjects can be coupled, e.g. for data science collaboration between care and research projects. The central concept is the *case*, which can be fully customised for a certain diagnosis. Users will have worklist for the tasks to be done at their group or department.

Generally, writing and viewing access of data is mostly authorised at the *case* level, analogous to medical practice. The *patient* or the *subject* is the identifier for the person to which the *case* belongs. Organising data in *cases* allows collaborations between medical departments, like pulmonology and oncology for cases with comorbidity of asthma and oncology, respectively. *Cases* may contain *consultations* to structure data in anamnesis, direct measurements, and questionnaires send (in)directly to the patient. *Cases* may also contain *test cascades* to collaborate between departments, like a pulmonology department requesting blood tests from haematology. These test cascades will orchestrate the appropriate rights to view and edit, as well as the workload list of various departments. *Test requests* are generated by the appealing departments and the *test results* and/or *test conclusions* are entered by the performing departments. Data management can be organised over the boundaries of any single organisation, as LogiqSuite is fully cloud based and can allow multiple organisations to collaborate under precise restrictions. This collaboration could be sharing data at the *case* level or within *test cascades*.

Descriptions in the user interface and data fields can be translated to the user's preferred language, e.g. to avoid translation of physician-patient interactions, and patient's language, e.g. in the case of questionnaires sent directly to patients. When entering choices of options, the user interface is switch to the user language (e.g. Dutch), while syncing the appropriate English translation to LogiqAnalytics for data science purposes.

### Solution design

The LogiqSuite application is open to external interactions to other applications, like clinical decision support tools (e.g., U-Prevent, which is also built by Ortec), and has open API for third party apps (Fig 2). Data communication can be done using *patient*, *subject*, *case*, or *test* identifiers.

Prediction models can be (f) productised for risk predictions for clinical-decision support, that are (g) coupled with the central cases. Central cases also allow (h) open API communication with third party apps.

### Data transfer access

Data from other databases can be mapped and imported into LogiqSuite by extract–transform–load (ETL) procedures. The ETL process allows verifiable, reliable, and repeatable data import from different sources into LogiqSuite. During the Transform step of ETL, data curation was also performed on the data from the various sources.

### Data science in LogiqAnalytics

LogiqAnalytics consists of a SQL database with numerical data and predefined choice options. All relevant data are real-time synced to LogiqAnalytics. Some data might enhance personal identifiability, like dates of birth and visits, and free text. In analytics the dates are converted to numerical data like age at diagnosis or duration of disease. Free texts are not suitable for analytics, so data intended to be analysed should be grouped into option prior data entry. The

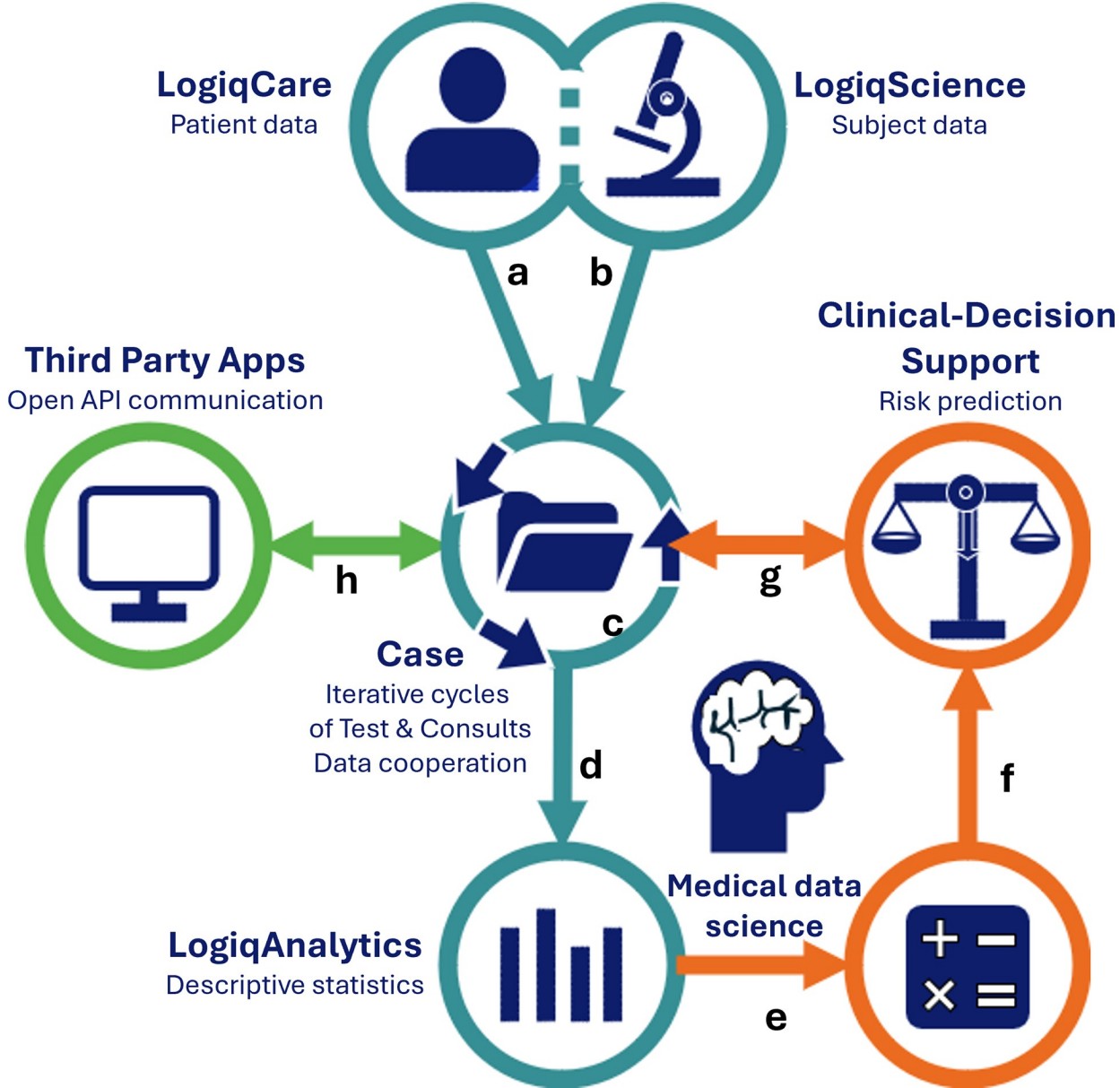

**Fig 2. LogiqSuite landscape.** LogiqSuite (blue) with interactions to other solutions from ORTEC (orange) and external sources (green). In brief, LogiqCare allows recording of (a) Patient data in LogiqCare, and (b) Subject data in LogiqScience, with (c) Cases as a central database. Case data are (d) synced to LogiqAnalytics for real-time descriptive statistics, and (e) ready-to-use for medical math, using AI to develop new prediction models.

database structure in LogiqAnalytics can be configured to the needs of the project using SQL view tables, e.g. mapped into a relevant data model structure. This table structure also allows different parties to see different domains of data.

Additionally, it supports proper metadata documentation by an integrated data dictionary. The deidentified data are available in real-time for descriptive analytics in generated reports with Microsoft Power BI, Excel, SPSS, and other applications. The data can also be used for advanced analytics and AI-powered model development.

## Results

Although biomedical research has some generic principles, it also has a shear infinite number of putative solutions. Implementation in different projects is preceded by customization and regulation of user access.

### Customisation process

DMSs for Personalized medicine should be flexible to allow the rapid creation of complex databases for collaboration of many scientific groups. Database design should be plastic and robust for progressive insights and historical consistency. Upon saving data, templates are used to build entities, that warrant data integrity independent of later database changes. The databases are configured using data templates for *cases*, *consultations*, and *test request*s, *test results*, and *test conclusions*. These are fully designable using standard building blocks in Logiq-Care and LogiqScience.

Users lead the customisation process by defining their templates in an Excel file, supported by a medical data scientist of the LogiqSuite team (Fig 3). The medical data scientists provide crucial input for data stewardship before data collection to facilitate real-time data analytics. After verification by a medical data scientist, the Excel template is converted to a Logiq template on the pilot environment. User interface of templates could be translated to any language e.g., for international studies. The templates function as data-entry screens in the pilot environment. Next, template validation is done by an appointed user. After customer approval, the template is made available on the production environment. Changes could be implemented rapidly, e.g. if needed within a day, while maintaining the careful multistep process.

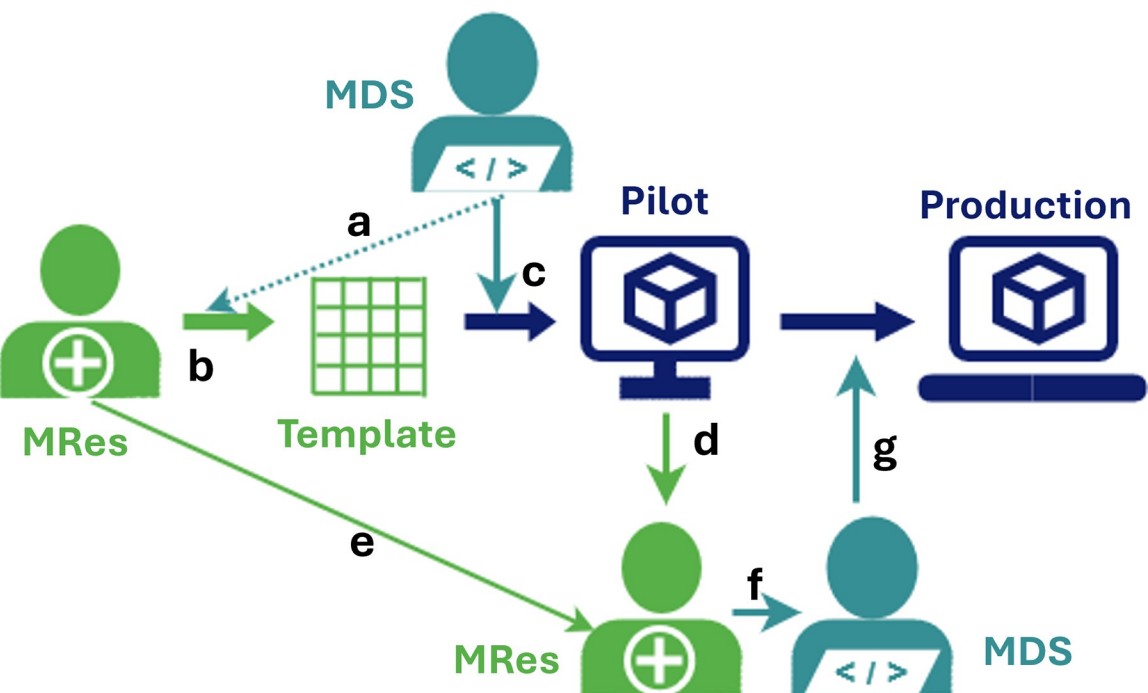

**Fig 3. Template customisation process.** In brief, (a) medical data scientist (MDS) provides a template to designated (b) medical researcher (MRes) who will design the desired data flows with MDS advice. When finalised, (c) the MDS will run the python script to upload the template in the Pilot environment, where (d) the medical researcher will validate the template, for (e) improvements or (f) to instruct the MDS to (g) transfer the template to the production environment.

In the LogiqAnalytics database data could be regrouped if desired for certain analytics, but in general the structure of the LogiqCare and LogiqScience template is used. Data can be enriched with a user defined data dictionary, to give meaning to values for analysis, and to facilitate FAIR data exchange.

## Regulation of user access

Access control is supervised using a documented four-eyes principle i.e., the person granting access is not the same person as executing access. Project leaders assign the persons to decide on role- (RBAC) and attribute-based access control (ABAC), reading and writing rights for a maximum of one year per request. The requests are logged automatically for traceability and executed manually by ORTEC to avoid abuse.

LogiqSuite provides its own Azure AD as Identity Provider (IdP) to which members of another organisation can be invited using federation identity. This implies that company policies for data access apply. Authenticating to the LogiqSuite Azure AD requires an additional MFA, that might be merged into the external Azure AD's authentication policy. Single sign-on is supported. For individual users, LogiqSuite can provide its own Azure AD as IdP, equipped with MFA that can be enforced.

User access is granted by the project lead and executed by ORTEC to obey the four-eyes principle. Only specific ORTEC users have roles with exclusive rights, like Admin, to manage accounts, departments, and groups, and Configurator, to edit studies and templates for data, e-mails, and reports. Roles for customer users are constrained to User and Viewer, limited by attributes for accounts, departments, and Care and/or Science. To avoid unintended interactions between functions, distinct RBAC and ABAC are separated for each task. After login, LogiqSuite users with multiple functions can switch between their active functions with distinct RBAC and ABAC, to avoid unintended control due to combining functions, e.g. an editing right in function A should not change a viewing right in function B.

## Evaluation of precision medicine use cases

LogiqSuite has three different privacy levels i.e., directly identifiable, pseudonymous, and (shear) anonymous for care, research (science), and analytics, respectively. This allows users to select their desired balance between protection of patient and data safety.

In seven different biomedical projects we could identify five different use-cases for precision medicine: (I) reorganisation and integration of data analytics for descriptive statistics in oncology, (II) real-time monitoring of a multicentre clinical study for pulmonary disease, (III) integration of clinical patient data with GMP data in oncology, (IV) AI development of for clinical decision support (CDS) models in oncology, and (V) integration of real-world data with CDS in cardiovascular risk management and prehospital triage.

## I. Integration of databases for analyses

Different databases contained parts of the clinical follow-up data on minimal residual disease (MRD) from Acute Myeloid Leukaemia (AML) [48]. In this research project, data was available in different databases from haemato-oncology of Adults in the Netherlands (HOVON) [49] and local databases built by the Amsterdam UMC, which had data on patient-survival follow-up and follow-up leukaemia-aberrant immune phenotypes (LAIPs), respectively (Fig 4). A common data model was crafted combining the structure of these databases in a subject-case structure with consults for direct analysis, and test cascades with test-requests for samples, test-results for direct analyses, and test-conclusions for LAIP-specific data.

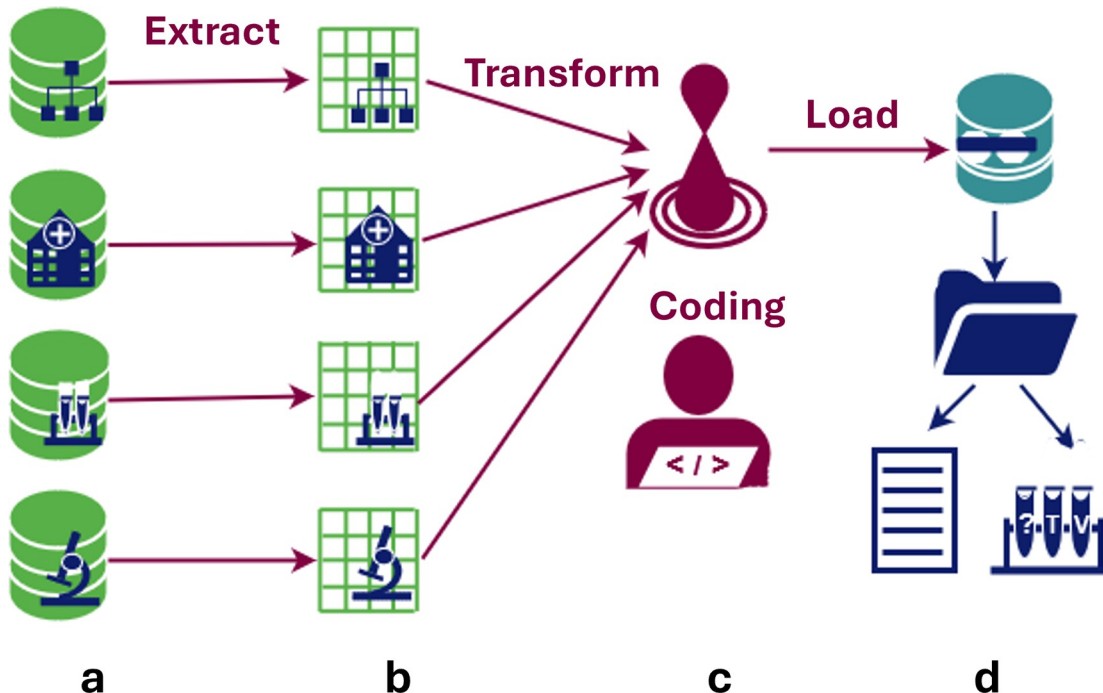

**Fig 4. Extract-Transform-Load integration for AML.** In brief, (a) data was extracted from different databases, which were coupled with unique identifiers, (b) data transformation consisted of mapping to the LogiqSuite's data structure and values, that were translated and curated before loading into LogiqSuite, (c) which are written and stored as F-sharp code, to allow tracking and reusing of the procedure, and (d) data are loaded in LogiqSuite and organized in Patient or Subject–Case–Consults & Test cascade structures.

A research MedDMS was built in LogiqScience, and the data were synced to LogiqAnalytics. The data consist of subject, the genetic characterisation of their AML, the characterisation, classification, and quantification of LAIPs, and therapy and survival data. Data was loaded using ETL from a csv file out of a query from the HOVON database and multiple local data management systems in Amsterdam UMC and GIMEMA in Rome.

## II. Real-time monitoring of a multilocation clinical study

Clinical studies in precision medicine are often performed at multiple locations in different languages, while they ideally should be monitored, orchestrated, and managed at a central location in real time. The results are real-time synced for data analytics. The LogiqScience orchestrates the clinical study as a central electronic case report form (eCRF) in LogiqScience. Different attributes were used for ABAC to limit data access on a need-to-know basis, e.g. by only providing access to subjects of a user's centre. Two additional institutes contributed data on lifestyle intervention and air pollution exposure, which had different roles limiting which subset of the Subject's data could be accessed. Additionally, Test cascades allowed cooperation between departments while properly controlling data access on a need-to-know basis.

The Long Covid use case [50] belongs to the precision medicine for lung diseases (P4O2) research project, a multi-site clinical study for lung diseases for precision medicine, including a Long Covid study. Data was entered by five different inclusion centres directly into LogiqScience as an eCRF. The eCRF templates used by the researchers were designed in English, but the questionnaire templates for the patients were available in Dutch. The multilingual properties of LogiqCare and LogiqScience offer these possibilities, but LogiqAnalytics is

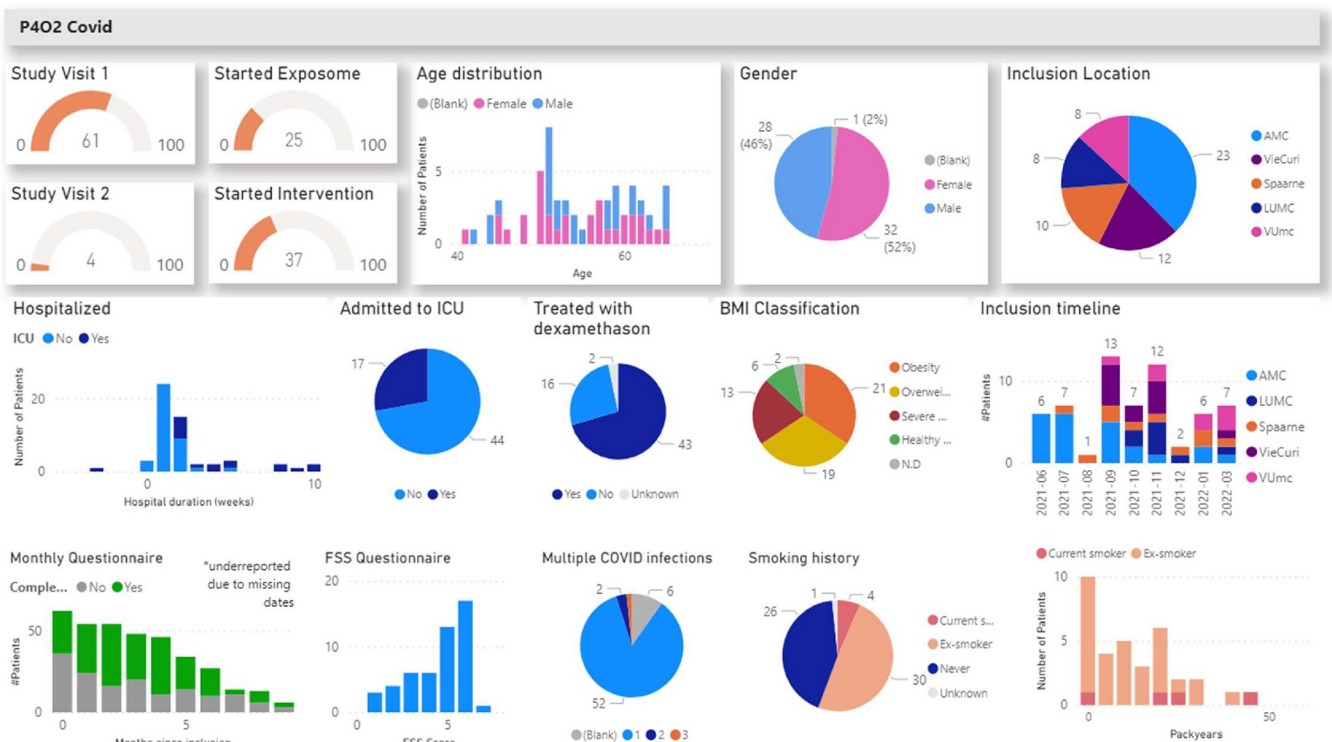

**Fig 5. Early study dashboard P4O2 Covid showing trial progress state in April 2022.** In brief, data are synced in analytics, showing real-time study overview. Arrows point to missing data, which are indicated in grey. This information was used for real-time identification of missing data, and early corrective actions.

synced in English only. Choices, dates, and numerical values were synced to LogiqAnalytics, allowing real-time monitoring of the progress in subject inclusion and data completeness by Power BI dashboards (Fig 5). Data on air pollution by ultrasonic personal aerosol sampler was transferred into the database by ETL. Researchers could monitor the data availability dashboard and provide real-time feedback to users, such that data could be completed in time. As a result of the real-time sync to LogiqAnalytics, as soon as the data of the last subject was entered into the eCRF, the analytics database was ready for analysis.

## III. Integration of care and precision medicine production

The AIDPATH project aims to set up AI-powered decentralised production of advanced (CAR-T cell) therapies in hospitals all over Europe. Production of precision medicine or advanced therapeutic medicinal products, like the production of autologous CAR-T cells, requires integration of clinical care and good manufacturing practice (GMP) production data for advanced therapies. Using LogiqSuite, it is possible to integrate a patient's clinical care data in LogiqCare with pseudonymised CAR-T cell production data in LogiqScience (Fig 6).

The AIDPATH project consortium aims to have AI-guided production of autologous chimeric antigen receptor T-cells (CAR-T cells) distributed over various UMCs [51]. In the AIDPATH project, two flows are combined, a clinical with patient data with a GMP production with subject data, allowing direct patient identification where needed, and preserving patient privacy where possible. Specific pretreatment patient data are crucial for CAR-T cell production and thus integrated in shared cases between patients and subjects. Some details are only needed for care or manufacturing purposes, so case data might be shared, but underlying consultations and test cascades could be unshared. This allows seamless collaboration between

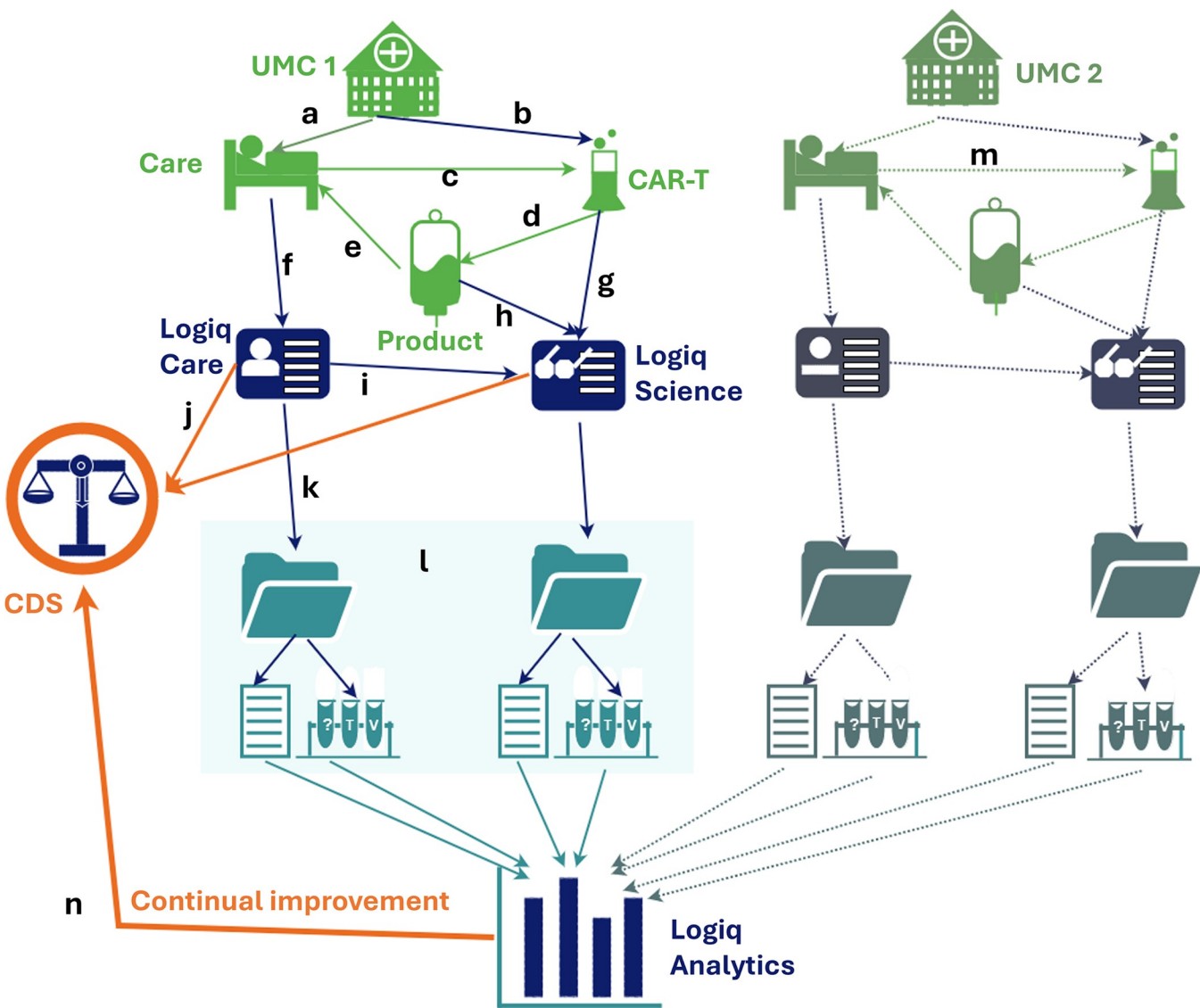

**Fig 6. Integration of care and GMP processes in AIDPATH.** In brief, UMC 1 has (a) care and (b) CAR-T cell productions of (c) autologous cells, coupling the products and (d) their quality control to the (e) patient treatment. Care will be (f) recorded in LogiqCare and (g) ATMP / GMP CAR-T cell production and (h) quality control in LogiqScience, which are (i) interconnected. If a clinical-decision tool is developed this can be (j) used to determine a desired CAR-T cell product from personal clinical data, and (k) provide this information to adapt the production process. Data within a case (l) contains consults and test cascades (Fig 4), (m) data from other institutes (UMC 2) is added to LogiqAnalytics and all data are gathered for (n) continual learning to improve the clinical-decision support tool.

care and manufacturing, ensures the pseudonymised production data are unambiguously coupled to the patient, without sacrificing patient privacy [52].

The decentralised AIDPATH production system uses laboratory devices to automatically process the CAR-T cells. The devices are digitally connected to the COPE (Control Optimise Plan Execute). COPE acts as a manufacturing execution system and connects to the devices using their respective software interfaces. This enables detailed live data logging, process control as well as online production parameter modification. The COPE Software acts as a user interface for the automation environment showing the pseudonymised patient data from the LoqiqSuite platform. Using the LogiqSuite platform, data from all decentralised production

sites will be registered in a standardised manner. The resulting LogiqAnalytics database will uniformly combine data from all sites and be crucial for continuous learning and model development.

## IV. AI development from a database

In two oncological projects, AML-MRD and AIDPATH, data from LogiqAnalytics is used for machine learning algorithms to develop predictive models for survival analysis with competing risks, that can be used as CDS tools [53]. The data was imported from other software solutions (e.g., Access-, SPSS-, and Excel-databases) using ETL procedures and integrated in LogiqScience. The data from AML-MRD has been described above (e.g. Fig 4). The data for the AIDPATH use case 5 was derived from multicentre clinical trial ARI-0001 [54,55] and ARI-0002h [56].

In the AML-MRD case the competing risks were graft versus host disease (GvHD) and toxicity (chemotherapy related risk), and tumour relapse. In the CAR-T cell case the competing risks were cytokine-release syndrome, immune suppression, and tumour relapse.

Data from LogiqScience was synced to LogiqAnalytics. Available and putative relevant biomedical parameters were selected, based on their potential mechanistical relevance and their timely availability for prognosis in practice. Machine learning was employed directly on data in LogiqAnalytics [57]. All these activities can be performed directly on the real-time synced data in LogiqAnalytics, allowing the CDS prediction model to be updated without having to rebuild the machine learning flow. Currently, we are working on the parameter selection and data analytics in these projects to develop a prediction model.

## V. Real-world data integration with clinical-decision support tools

ORTEC has developed and connected two CDS tools, U-Prevent and the Stroke Triage App, to LogiqScience to collect data directly after CDS use. Since LogiqAnalytics is fed real-time from LogiqCare and LogiqScience, this allows continuous learning of the CDS prediction model. After expert evaluation for MDR this results in continual learning to improve the AI CDS tools.

U-Prevent is an MDD-certified clinical decision support solution with multiple prediction models for cardiovascular risk management. Practical use is greatly enhanced by using data directly from electronic health records [58]. U-Prevent allows users to enter missing data or to use imputation [59]. In the recent PROSPERA project, we have coupled LogiqSuite to U-Prevent (see Fig 2), allowing it to prefill data in U-Prevent from the primary care electronic health records in U-Prevent. After data revision and updates by the physician, the U-Prevent CDS tool can be used in patient care to support share decision making and allocating appropriate care. At the end of the U-Prevent session, the updated data and CDS outcomes are stored in LogiqScience. Hereby, LogiqSuite facilitates efficient use of a clinical decision support tool and automatic registration of follow-up data, in this case in a pseudonym way due to local interpretations of the GDPR. Moreover, the analytics environment yields real-time dashboard with included patients, genders, risks profiles, prevention strategies, and more, that can easily be selected per primary care centre. These key-performance indicators (KPIs) facilitate benchmarking of various practitioners.

For prehospital triage of patients with suspected acute stroke we developed the Stroke Triage App, an MDR class I CDS tool, based on the risk predictions and timetables of PRESTO-2 implementation study [60,61] and disease history and routing information. In brief, the Stroke Triage App supports routing decisions for emergency care by advising on the adequate hospital level for stroke care in relation to the patient's prehospital triage assessment. Pseudonymous

patient identifiers, data from prehospital triage, GPS location, routing information, and advised outcome are stored in LogiqScience. During the study the data can be enriched with clinical follow-up data. This will allow the continual evaluation of the Stroke Triage App and facilitate continuous improvement of the underlying CDS prediction model for prognosis used in routing decisions.

## Discussion

Precision medicine stratifies diagnosis-treatment combinations between similar diseases, using large and complex datasets. This implies the need for good quality large databases of complex medical data that should be GDPR compliant, collect real-world data, integrate care and research, allow FAIR data exchange with multiple parties, communicate with clinical-decision support tools, easy to adapt for progressive insight, facilitate real-time analysis, and provide ready-to-use data for AI methods to develop new prediction models for prognostic, diagnostic and therapeutic purposes. In different use-cases we have described a MedDMS system that fulfils the basic criteria. The different approach of the integrated solution does not end but opens important subjects of discussion.

### Data protection

MedDMSs for precision medicine goes beyond data sharing in the cloud and demands a higher level of traceable procedures. This includes documentation of implementation, data stewardship for the traceability of design modification, data validation steps, and privacy protection by a fine-grained authorisation network. The automatic recorded and verified process documentation becomes more important where research complexity grows in number of parameters and inclusion centres. The plan for data management is crucial, so a MedDMS cannot seen independent from its procedures to maintain confidentiality of data and consistency of the database.

LogiqSuite allows integration of medical care and clinical research with real-time available descriptive statistics, and machine learning in one solution, with different and appropriate privacy levels for each use case. LogiqSuite distinguishes itself from other MedDMS by its interactions with CDS prediction models for precision medicine as shown in the two double use-cases. LogiqSuite has prepares real-world data in real-time for development. Data from Logiq-Suite is directly available in CDS tools, facilitating their easy use. This is linked to the feedback of the (modified) input and results into LogiqSuite. Syncing these data in real-time to LogiqAnalytics facilitates the cycle of continual learning for precision medicine.

Sharing data from different study sites applies standard and interoperable solutions for implementing and managing medical registries to each site [62]. LogiqSuite allows this with the same data templates, but ABAC restricts data access to those authorised to the relevant attributes e.g., institutes. Input validation is an important asset of data stewardship [40]. The medical data scientists guide the use of these and other features in the MedDMS implementation.

### Comparing MedDMSs

LogiqSuite is not the first cloud MedDMS. REDCap exists since 2009 [63] and is used for multilocation eCRFs [64]. Castor EDC [65] and FIMED [66] also support storage of medical research data. Castor EDC has a signable integrated informed consent for research purposes, which should technically be a trusted third party (TTP) i.e., with separate access authentication. LogiqSuite distinguishes from other MedDMS using four-eyes principles to guard data structure, quality, and access. Since this process loops in the medical data scientists, they can

also give advice for dynamic templates and live analytics, which are new for many medical researchers. LogiqSuite also has added template translations, sending of questionaries to patients or subjects, connectivity with open API, and support with data transfer from other sources by ETL. LogiqSuite users are charmed by the real-time study monitoring, which is suitable for monitoring study completeness and exerting needed corrective actions.

## Integrated versus federated MedDMS

Not all involved in precision medicine choose to integrate MedDMSs in a single solution. Some parties propose blockchain with communication tools between physically separate MedDMSs [67]. Separate MedDMSs offers some easy advantages for privacy protection, but ABAC and RBAC strategies on a need-to-know basis could also do this. Federated learning methods in healthcare are developed to allow machine learning to be exerted. The major challenge in machine learning is the poor data quality yielding poor CDS models. While federated learning is pioneered in various projects, methods for data stewardship and curation in federated learning still needs to be explored.

Separate MedDMSs will have different interpretations of parameters and different data quality. Poor quality data could be curated, which is a critical step in machine learning [68]. Input validation and data curation techniques might be suboptimal for federated approaches [69,70]. Options are to perform this either centrally [71] or by general rules (e.g. x times the standard deviation) [72]. In our experience, these methods have limited efficacy for curing complex, large, divergent medical data with missing properties. Other methods for data curation are needed for efficient use of federated learning techniques. Especially, systematic shifts due to confounders between data subgroups should be detected and understood. Ideally data stewardship assesses an anonymised data set completely, since this facilitates the application of the complex rules for data stewardship more appropriately.

## Beyond the MedDMS

Even if many medical research groups collaborate in single database, there will always be medical data beyond the current collaborations, which also needs to be integrated for scientific research. So MedDMSs should be prepared for FAIR data exchange. Medical data are complex by their nature, consisting of various domains (e.g. anamnesis, follow-up, disease classifications, lab tests, medication) and includes various types of data (e.g. dates, values, repeated measurements), which use various methods for data classifications. After early and continued initiatives for Observational Medical Outcomes Partnership (OMOP) common data model [73,74], the scientists expanded this to collaborate in different settings for FAIR data exchange [75]. Common data models are important for various purposes and crucial for AI [76]. Systems are being developed to add FAIR standards de novo to MedDMSs [77]. Multiple initiatives exist to draw data details of FAIR sharing [78], many in national or regional settings. Combining complexity of medical data and the regional approach, we foresee that there will be multiple relevant common data models and corresponding data dictionaries. The LogiqAnalytics data can be filled with all relevant predefined common data model structures, using SQL views. In LogiqSuite multiple data dictionaries can be integrated for FAIR data exchange.

## Conclusions

LogiqSuite is a MedDMS that integrates directly identifiable care data, pseudonymous science data and minimal identifiable to anonymous analytic data. LogiqSuite was evaluated in five different use cases for precision medicine, including data research, multilocation study monitoring, integration of research and production data with care data, real-time use of data for

prediction model development, and input and registration of data from CDS tools. The value of this MedDMS was shown in different biomedical fields, including oncology, cardiovascular risk management, pulmonology, and prehospital triage.

LogiqSuite is unique in supporting real-time data analysis in care settings, which is also a beloved feature for scientific researchers. Moreover, LogiqSuite supports collaborative clinical care and research/laboratory workflows. The data can originate from manual entry in clinical care, eCRFs of research studies, clinical-decision support tools, import via ETL from third-party sources, and through FAIR-compliant open API. Available data can be monitored in real-time, providing tools for data monitoring and continuous feedback loop for data analytics and prediction model development. In clinical practice this facilitates continual learning with real-world data, since MDR legislation require a clinical evaluation for the validity of updated prediction models in every cycle.

Appropriate tools are crucial for KPI monitoring and progress in care and science. The LogiqSuite MedDMS application supports data management and science for advancement to precision medicine by data collection and collaboration, facilitating analytics and machine learning, and implementation of CDS tools.

## Acknowledgments

The authors would like to thank Bob Roozenbeek and Ruben van Wijdeven (Erasmus MC, Rotterdam, The Netherlands) for their invaluable contribution to the Stroke Triage App. The authors would like to thank Sandy Pratama, Thom Steenhuis, Marco Koeleman, Christian Hutter, Vlad Constantin Lipan and Alina Bratosin for their excellent technical assistance on the development of LogiqSuite.

## Author Contributions

**Conceptualization:** John J. L. Jacobs, Jacqueline Cloos, Steven van Dijk.

**Data curation:** John J. L. Jacobs, Inés Beekers, Inge Verkouter, Levi B. Richards, Lok Lam Ngai, Jesse Tettero.

**Formal analysis:** John J. L. Jacobs, Inés Beekers, Inge Verkouter, Levi B. Richards, Alexandra Vegelien, Simon Hort.

**Funding acquisition:** John J. L. Jacobs, Anke H. Maitland-van der Zee, Qasim A. Rafiq.

**Investigation:** John J. L. Jacobs, Inés Beekers, Inge Verkouter, Levi B. Richards, Alexandra Vegelien, Lizan D. Bloemsma, Vera A. M. C. Bongaerts, Jacqueline Cloos, Frederik Erkens, Simon Hort, Michael Hudecek, Manel Juan, Anke H. Maitland-van der Zee, Sergio Navarro-Velázquez, Lok Lam Ngai, Qasim A. Rafiq, Carmen Sanges, Jesse Tettero, Hendrikus J. A. van Os, Rimke C. Vos, Steven van Dijk.

**Methodology:** John J. L. Jacobs, Inés Beekers, Inge Verkouter, Levi B. Richards, Alexandra Vegelien, Lizan D. Bloemsma, Vera A. M. C. Bongaerts, Jacqueline Cloos, Frederik Erkens, Simon Hort, Michael Hudecek, Manel Juan, Anke H. Maitland-van der Zee, Sergio Navarro-Velázquez, Lok Lam Ngai, Qasim A. Rafiq, Carmen Sanges, Jesse Tettero, Hendrikus J. A. van Os, Rimke C. Vos, Yolanda de Wit, Steven van Dijk.

**Project administration:** John J. L. Jacobs, Jacqueline Cloos, Patrycja Gradowska, Simon Hort, Michael Hudecek, Manel Juan, Anke H. Maitland-van der Zee, Qasim A. Rafiq, Hendrikus J. A. van Os, Yolanda de Wit.

**Resources:** John J. L. Jacobs, Inés Beekers.

**Software:** John J. L. Jacobs, Inés Beekers, Steven van Dijk.

**Supervision:** John J. L. Jacobs.

**Validation:** John J. L. Jacobs, Inés Beekers, Inge Verkouter, Steven van Dijk.

**Visualization:** John J. L. Jacobs, Inés Beekers, Inge Verkouter.

**Writing – original draft:** John J. L. Jacobs.

**Writing – review & editing:** John J. L. Jacobs, Inés Beekers, Inge Verkouter, Levi B. Richards, Alexandra Vegelien, Lizan D. Bloemsma, Vera A. M. C. Bongaerts, Jacqueline Cloos, Frederik Erkens, Patrycja Gradowska, Simon Hort, Manel Juan, Sergio Navarro-Velázquez, Carmen Sanges, Hendrikus J. A. van Os, Rimke C. Vos, Yolanda de Wit, Steven van Dijk.

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
