## [Decision Letter · Decision Letter 0]

26 Mar 2024

PDIG-D-24-00057

A data management system for precision medicine

PLOS Digital Health

Dear Dr. Jacobs,

Thank you for submitting your manuscript to PLOS Digital Health. After careful consideration, we feel that it has merit but does not fully meet PLOS Digital Health's publication criteria as it currently stands. Therefore, we invite you to submit a revised version of the manuscript that addresses the points raised during the review process.

Please submit your revised manuscript within 30 days Apr 25 2024 11:59PM. If you will need more time than this to complete your revisions, please reply to this message or contact the journal office at digitalhealth@plos.org. Please include the following items when submitting your revised manuscript:

We look forward to receiving your revised manuscript.

Kind regards,

Mengyu Wang, Ph.D.

Academic Editor

PLOS Digital Health

Journal Requirements:

2. Please send a completed 'Competing Interests' statement, including any COIs declared by your co-authors. If you have no competing interests to declare, please state "The authors have declared that no competing interests exist". Otherwise please declare all competing interests beginning with the statement "I have read the journal's policy and the authors of this manuscript have the following competing interests:"

3. Please provide separate figure files in .tif or .eps format only and remove any figures embedded in your manuscript file. Please also ensure that all files are under our size limit of 10MB.

co

4. Please amend your detailed Financial Disclosure statement. This is published with the article. It must therefore be completed in full sentences and contain the exact wording you wish to be published.

If you did not receive any funding for this study, please simply state: “The authors received no specific funding for this work.

5. Please provide a complete Data Availability Statement in the submission form, ensuring you include all necessary access information or a reason for why you are unable to make your data freely accessible. If your research concerns only data provided within your submission, please write "All data are in the manuscript and/or supporting information files" as your Data Availability Statement.

Additional Editor Comments (if provided):

Reviewers' comments:

Reviewer's Responses to Questions

**Comments to the Author**

1. Does this manuscript meet PLOS Digital Health’s publication criteria? Is the manuscript technically sound, and do the data support the conclusions? The manuscript must describe methodologically and ethically rigorous research with conclusions that are appropriately drawn based on the data presented.

Reviewer #1: Yes

2. Has the statistical analysis been performed appropriately and rigorously?

Reviewer #1: Yes

3. Have the authors made all data underlying the findings in their manuscript fully available (please refer to the Data Availability Statement at the start of the manuscript PDF file)?

Reviewer #1: No

4. Is the manuscript presented in an intelligible fashion and written in standard English?

Reviewer #1: Yes

5. Review Comments to the Author

Reviewer #1: "This paper evaluates a MedDMS in five types of use cases for precision medicine, ranging from data

40 collection to algorithm development and from implementation to integration with real-world data.

41 The MedDMS is evaluated in seven precision medicine data science projects in prehospital triage,

42 cardiovascular disease, pulmonology, and oncology.", however the there is not a clearity about the availability of the data used to test MedDMS, or any restriction it has should be on the Data Availability Statement at the start of the manuscript PDF file.

6. PLOS authors have the option to publish the peer review history of their article (what does this mean?). If published, this will include your full peer review and any attached files.

**Do you want your identity to be public for this peer review?** For information about this choice, including consent withdrawal, please see our Privacy Policy.

Reviewer #1: No

---

## [Editor Report · Decision Letter 1]

3 Jul 2024

PDIG-D-24-00057R1

A data management system for precision medicine

PLOS Digital Health

Dear Dr. John J. L. Jacobs,

Thank you for submitting your manuscript to PLOS Digital Health. After careful consideration, we feel that it has merit but does not fully meet PLOS Digital Health's publication criteria as it currently stands. Therefore, we invite you to submit a revised version of the manuscript that addresses the points raised during the review process.

Please submit your revised manuscript within 30 days on 02 August 2024. If you will need more time than this to complete your revisions, please reply to this message or contact the journal office at digitalhealth@plos.org. Please include the following items when submitting your revised manuscript:

We look forward to receiving your revised manuscript.

Kind regards,

Raymond Francis Sarmiento, MD

Section Editor, Implementation Science for Digital Health and Medical A.I.

PLOS Digital Health
---

## [Editor Report · Decision Letter 2]

8 Oct 2024

A data management system for precision medicine

PDIG-D-24-00057R2

Dear Dr Jacobs,

We are pleased to inform you that your manuscript 'A data management system for precision medicine' has been provisionally accepted for publication in PLOS Digital Health.

Best regards,

Jennifer N Avari Silva, MD

Section Editor

PLOS Digital Health

Thank you for your response to previous comments.